# Tackling Intrinsic Antibiotic Resistance in *Serratia marcescens* with a Combination of Ampicillin/Sulbactam and Phage SALSA

**DOI:** 10.3390/antibiotics9070371

**Published:** 2020-07-01

**Authors:** Lorenz Weber, Mathias Jansen, Alex Krüttgen, Eva Miriam Buhl, Hans-Peter Horz

**Affiliations:** 1Institute of Medical Microbiology, RWTH Aachen University Hospital, 52074 Aachen, Germany; lorenz.weber@rwth-aachen.de (L.W.); mathias.jansen@rwth-aachen.de (M.J.); 2Laboratory Diagnostic Center, RWTH Aachen University Hospital, 52074 Aachen, Germany; akruettgen@ukaachen.de; 3Electron Microscopy Facility, RWTH Aachen University Hospital, 52074 Aachen, Germany; ebuhl@ukaachen.de

**Keywords:** *Serratia marcescens*, intrinsic antibiotic resistance, bacteriophage, synergy

## Abstract

During the antibiotic crisis, bacteriophages (briefly phages) are increasingly considered as potential antimicrobial pillars for the treatment of infectious diseases. Apart from acquired drug resistance, treatment options are additionally hampered by intrinsic, chromosomal-encoded resistance. For instance, the chromosomal ampC gene encoding for the AmpC-type β-lactamases is typically present in a number of nosocomial pathogens, including *S. marcescens*. In this study, phage SALSA (vB_SmaP-SALSA), with lytic activity against clinical isolates of *S. marcescens*, was isolated from effluent. Besides phage characterization, the aim of this study was to evaluate whether a synergistic effect between the antibiotic ampicillin/sulbactam (SAM) and phage can be achieved despite intrinsic drug resistance. Phage SALSA belongs to the *Podoviridae* family and genome-wide treeing analysis groups this phage within the phylogenetic radiation of T7-like viruses. The genome of Phage SALSA consists of 39,933 bp, which encode for 49 open reading frames. Phage SALSA was able to productively lyse 5 out of 20 clinical isolates (25%). A bacterial challenge with phage alone in liquid medium revealed that an initial strong bacterial decline was followed by bacterial re-growth, indicating the emergence of phage resistance. In contrast, the combination of SAM and phage, together at various concentrations, caused a complete bacterial eradication, confirmed by absorbance measurements and the absence of colony forming units after plating. The data show that it is principally possible to tackle the axiomatic condition of intrinsic drug resistance with a dual antimicrobial approach, which could be extended to other clinically relevant bacteria.

## 1. Introduction

In light of the inevitable spread of multi-drug resistance among clinically relevant bacterial species, new ways of antibacterial therapy are desperately needed. This urgency has led to a reconsideration of the therapeutic use of bacteriophages (briefly: phages) [1,2,3]. A number of desirable features are associated with phages, e.g., protection of the microbiome and target-dependent phage multiplication. However, one serious problem is the emergence of phage-resistant variants which could potentially jeopardize the success of phage therapy [4,5]. Recent studies have shown that the combined application of phages together with antibiotics not only leads to a stronger antibacterial activity than either substance alone, but also reduces the chances for the occurrence of phage resistant variants [6]. Importantly, synergistic interactions have been observed even with an antibiotic against which the targeted bacterial pathogen is *a priori* resistant [7]. One prerequisite for a successful dual antimicrobial approach is that two different selective pressures act upon the bacterial pathogen. Any mutational/phenotypical effort to escape the deleterious effects of one antimicrobial can then lead to increased sensitivity to the other [8].

The re-establishment of an ineffective antibiotic with a phage could expand treatment options, and allow clinicians to make use of familiar drugs, with well-known side effects and pharmacokinetics. In addition, even the apocalyptic scenario of pandrug resistance would leave actionable antagonizing opportunities for phage/antibiotic partnerships, as long as the bacterial pathogen is susceptible to the phage.

Apart from acquired drug resistance, treatment options are also frequently hampered by intrinsic, chromosomal-encoded resistance. For instance, the chromosomal ampC gene encoding for the AmpC-type β-lactamases is typically present in the so called “SPACE” group which includes *Serratia*, *Pseudomonas*, *Acinetobacter*, *Citrobacter*, and *Enterobacter* [9]. Unlike the other members of the SPACE-group, the clinical importance of *Serratia marcescens* has been underestimated and this species is considered an important human pathogen only in recent years. *S. marcescens* causes nosocomial infections and outbreaks in severely immunocompromised or critically ill patients, particularly in neonatal intensive care units (NICUs) [10,11]. Clinical manifestations are widespread and range from keratitis, conjunctivitis, urinary tract infections, pneumonia, surgical wound infections to sepsis, bloodstream infection and meningitis [10,12,13,14]. In NICUs ampicillin, the first “broad spectrum” penicillin, is of prominent use, as it is effective against a potpourri of different infectious diseases and bacterial species including anaerobes [15]. On top of this, ampicillin is cost-effective and well-tolerated by most patients. However, ampicillin is contra-indicated against infections caused by *S. marcescens*, due to its carriage of the chromosomal AmpC gene. The aim of the current study was to test the hypothesis, that the intrinsic ampicillin resistance in *S. marcescens* can be overcome with the co-addition of a lytic phage due to synergistic interactions. To this end, a natural phage, vB_SmaP_SALSA (briefly referred to as phage SALSA) was isolated from a sewage sample based on a clinical isolate of *S. marcescens* as propagation host. Besides the phenotypic and genomic characterization of this new phage, we performed liquid infections assays and evaluated the extent of bacterial reduction upon antibacterial challenges with varying dosage of drug or phage alone, in comparison with the combination of both compounds.

## 2. Results and Discussion

### 2.1. Characterization of S. marcescens Clinical Isolates

A total of 20 clinical isolates of *S. marcescens* (designated as SM01-SM20) obtained from various clinical specimen at the University Hospital RWTH Aachen, Germany in the year 2017, were used in this study. MALDI-TOF mass spectrometry verified all 20 clinical isolates as *S. marcescens*. While, otherwise exhibiting consistent resistance profiles, variable sensitivities for the antibiotics cefpodoxime, imipenem, moxifloxacin, and tigecycline, allowed the categorization of the 20 isolates into eight groups (Appendix A). A further differentiation of the isolates was possible via enterobacterial repetitive intergenic consensus PCR (ERIC-PCR), which ruled out the existence of duplicates (Appendix A).

### 2.2. Phenotypic Characterization of vB_SmaP-SALSA

Phage SALSA was isolated based on the clinical isolate SM01 from sewage obtained from the wastewater treatment plant Aachen-Soers, Germany. When spotted undiluted, phage SALSA formed zones of clearing against seven out of 20 (35%) different clinical *S. marcescens* isolates, i.e. SM01, SM04, SM10, SM11, SM14, SM19, and SM20 (Appendix A). Upon further dilution, the phage was able to lyse five out of 20 (25%) clinical isolates (SM01, SM04, SM10, SM11, and SM14), showing distinct plaques within an average size of around 1.6 mm. In contrast to the latter four isolates, which exhibited identical resistance profiles, SM01 was additionally resistant against cefpodoxim and was intermediate resistant against tigecycline. Setting the efficiency of plating (EOP) for the propagation host SM01 to 100% led to EOPs of 61%, 39%, 20%, and 26%, respectively, for the other four strains mentioned above. The fact that the strains SM19 and SM20 were lysed only with a high phage titer likely means, that the phage can adsorb to the bacteria, but cannot initiate a true infection. The adsorption of a very high number of phages then causes stress to the bacteria leading to cell death. However, the precise mechanism that acts upon those two strains, which could for instance, be “lysis from without” [16], remains unclear and would require further experimentation. A comparably host range, as observed with phage SALSA, was also observed by a recently published T4-like *Serratia* phage PS2 [17]. Remarkably, two other *Serratia* phages, namely vB_SmaA_2050H1 and vB_SmaM_2050HW, each belonging to different phage families and unrelated to our phage SALSA, were reported to infect 66.67% and 80% of different *S. marcescens* strains, respectively [18]. Narrow host ranges are typical of most phages. However, for phage therapy, this problem can be overcome by using a phage cocktail rather than the use of single phages. While it seems possible but also cumbersome to find natural phages with a broader host range, recent advances in synthetic microbiology have shown that the host range of T7-like phages can be extended by genetic engineering [19]. The technique is based on the design of hybrid particles displaying various phage tail/tail fiber proteins with which potentially any desired bacterial host can be targeted. Besides the use of phage cocktails, it is likely that tailored phages (e.g., modification of natural phages) will be one of the clues towards realization of phage therapy [20]. Nevertheless, owing to the immense diversity of phages in nature, it will always be worthwhile to continue “phage culturomics” [21], in order to enlarge our picture of the underexplored world of phages [22].

Phage SALSA shows an icosahedral head with a short, non-contractile tail (Figure 1). Based on this morphology it can be assigned to the *Podoviridae* family morphotype C1 [23]. Virions of phage SALSA have an average width of around 59 ± 2 nm and an average length of 65 ± 2 nm.

### 2.3. Genome Analysis of Phage SALSA

The genome of phage SALSA consists of 39,933 bp and has a GC-content of 50.54%. Comparison with publicly available phage genomes (megablast) revealed 22 significant alignment scores belonging to T7-like viruses of the *Podoviridae* family. Fifteen out of those 22 phage genomes are direct submissions without further characterization, except for the indication of the bacterial hosts which belong to various members of the *Enterobacteriaceae*. Phage SALSA showed highest sequence similarity with the genome of the *Serratia* phage SM9-3Y [24] with an nucleotide identity of around 96.4% and a coverage of 95%, Table 1.

Genome-based treeing analysis using VICTOR [25] grouped phage SALSA together with other *Enterobacteriaceae*-phages within the phylogenetic radiation of *Yersinia* phage phiYeO3-12 [26] and *Salmonella* phage phiSG-JL2 [27], which in turn are related to *Escherichia* phage T7 (Table 1, Figure 2). The analysis also clustered phage SALSA and *Serratia* phage SM9-3Y as belonging to the same species. However, when applying other algorithm formulas (i.e., D0 and D4) using VICTOR, phage SALSA was clustered as a separate species (data not shown).

The genome of phage SALSA encodes for 49 open reading frames (ORFs), of which 94% are transcribed from the same DNA strand (Appendix A). Pairwise comparison of translated ORFs with the five phage genomes in Table 1 revealed that most ORFs (22 to 26 ORFs) share a 96 to 99% sequence identity at protein level (Appendix A). There are 4 to 9 shared ORFs with 100% sequence identity, all of which encode for hypothetical phage-like proteins. Shared ORFs with descending sequence identities in the range of 90 to 95%, 80 to 89%, 70 to 79%, 50 to 69%, and 20 to 49%, respectively, ranged between 1 and 8 ORFs. Phage SALSA contains 2 to 6 ORFs that are not present in the other phage genomes (Appendix A).

Functions could be assigned to 29 genes (59.2%) based on the genetic relatedness of the predicted proteins with the well annotated *Yersinia* phage phiYeO3-12 [26], Appendix A. Eighteen ORFs (36.7%) encode for proteins involved in replication, maturation, and release of phage progenies, 11 ORFs (22.4%) encode for structural phage proteins, while 20 ORFs (40.8%) encode for hypothetical phage-like proteins with yet unknown functions (Appendix A). The genome does not encode for any tRNA, which is in line with other T7-like viruses. Virulence genes, as well as attachment sites or ORFs encoding for integrase enzymes were not found in the genome of SALSA, indicating its lytic lifestyle.

### 2.4. Diversity of Known Serratia Phages

While a relatively high number of different *Serratia* phages have been isolated in the past [28], only four Serratia phage genomes were available in the NCBI viral genome database until recently [18,29]. However, as of May 2020, this number has increased to more than 20 phage genomes, where mostly *S. marcescens* is indicated as host. This comprises the six *Myoviridae* phages, “MyoSmar” [30], “Moabite” [31], “MTx” [32], “Muldoon” [33], PS2 [17], vB_SmaM_2050HW [18], and “BF” [29], the four *Siphoviridae* phages “Slocum” [34], “Serbin” [35], “Scapp” [36], and “Eta” [28], as well as the three *Podoviridae*-phages “Pila” [37], “Parlo” [38], and “SM9-3Y” [24]. Furthermore, the genome database contains the *S. marcescens* phage vB_SmaM_2050HW1 which belongs to the new family *Ackermannviridae* [18]. This rapid increase in comparably short time likely indicates the emerging interest in finding therapeutic useful phages against *S. marcescens* as a consequence of its increasing clinical importance. Additional phages in the genome database are *Myoviridae* phage phiMAM1 infecting *Serratia plymuthica*, [39], *Myoviridae* phage vB_Sru IME250 infecting *Serratia rubidaea*, [40], and *Myoviridae* phage PCH45 infecting an unspeciated member of the genus *Serratia*: strain sp. ATCC 39006 [41]. Lastly, the genome of three further *Myoviridae* phages, CHI14, X20, and CBH8 also infecting *Serratia* sp. ATCC 39006 are available in the database [42].

Another web-based resource (“Bacteriophage Names 2000”) provides additional insights into the diversity of isolated *Serratia* phages based on electron microscopy studies performed for several decades until the year 2000 (http://www.phage.org/names/2000/). Overall, this platform comprises the morphological classification of around 5000 phages infecting *Bacteria* and *Archaea* [23]. There are 85 *Serratia* phages listed in this database, of which 51 belong to *Siphoviridae*, 22 to *Myoviridae*, and 12 to *Podoviridae*.

Combining these frequencies together with the aforementioned and more recent phages present in the genome database (interestingly the frequency of *Myoviridae* is three times higher than in *Siphoviridae*) adds to a total of 34 *Myoviridae*-phages (33%), 55 *Siphoviridae*-phages (53%), but only 15 *Podoviridae*-phages (14%). Hence, with “Podo-phage” SALSA our work provides some data on a rather underrepresented phage family within the *Caudovirales*. To the best of our knowledge, this is the first study showing the antibacterial activity of a *Serratia* phage together with antibiotics.

### 2.5. Liquid Infection Assays

Liquid infection assays were performed with every strain that was permissive for productive lysis on solid medium and with the strains where the phage caused “lysis from without” [16]. This study focused on the investigation of possible positive interactions between phage SALSA and ampicillin/sulbactam (SAM) but included for comparative reasons also meropenem (MEM). All strains are intrinsically resistant against SAM but sensitive to MEM. Bacteria were challenged for 68h as planktonic cultures with either phage and/or the antibiotics SAM and MEM. Phage SALSA alone led to a rapid decline of the strain SM01 within the first 12h with both phage concentrations of MOI 1 and MOI 10^−2^ (red lines Figure 3). After this maximum peak of suppression, bacterial re-growth occurred steadily until a plateau was reached at around 30h, after which the bacterial growth slightly declined. The number of CFU/µL in the phage-alone assay after 68h was 3.0 × 10^4^ (MOI 10^−2^), and 1.0 × 10^5^ (MOI 1), respectively, Figure 3.

With the sole applications of the antibiotics SAM and MEM (blue lines) bacteria were initially able to grow until around 12 h after which a moderate bacterial decline occurred until 35h (MEM), and 44 h (SAM), respectively, followed by a slight bacterial increase until the end of the infection assay, Figure 3. The number of CFU/µL in the antibiotics-alone assays after 68h was 6.5 × 10^4^ (SAM), and 3.4 × 10^4^ (MEM), respectively, thus, comparable with the sole applications of the phage. One could argue that the phage-alone approach was not any more successful than the antibiotic-alone approach, given the similar magnitude of viable bacteria after 68 h. However, except for the phage attack, the favorable conditions of a full nutrient solution might enable the bacteria to exert whatever resistance mechanisms they dispose of. In contrast, under in vivo conditions, an initial phage-mediated decline might be accompanied by supportive action of the animal/human immune system, as demonstrated previously [43]. Therefore, there is some hope that phage resistance emergence plays a less important role during phage therapy [44,45]. Nevertheless, as more experimental data are needed to evaluate the universality of a collaboration between the immune system and phages, bacterial re-growth should always be a matter of concern. In order to overcome the phage attack, bacterial populations can either be characterized by genetic heterogeneity, where allelic variation is already present in few cells which gain a selective advantage after phage predation [46,47]. Alternatively, heterogeneity exists due to phenotypic differences (i.e., differential gene expression) within a clonal, genetically identical population [48]. Irrespective of the precise mechanisms, we frequently observed re-growth in our experiments especially when phages against *Enterobacteriaceae*-species were applied. Conversely, at least in our laboratory, bacterial re-growth has been less frequently observed with phages applied against other opportunistic pathogens, such as *Acinetobacter baumannii* [7], *Pseudomonas aeruginosa* [49] or *Staphylococcus aureus* (unpublished data). It appears that *Enterobacteriaceae* are particularly well-“armed” against phages [50]. Therapeutic success may therefore depend on the use of phage cocktails which prevents or at least retards phage resistance development.

Irrespective of this, in our experiments, the combination of phage and antibiotic drastically changed the extent of bacterial control (green lines, Figure 3). Bacterial reduction occurred as rapidly as in the phage alone. However, this time, the re-growth was suppressed and the absorbance curves stayed constantly at lowest position until the end of the infection assay. No CFUs were recovered after the 68h incubation period confirming complete bacterial clearance, Figure 3. Apparently, with both antimicrobials together, the bacteria where rigorously killed, so that any resistance response whether based on genetic or phenotypic mechanisms, was prevented. To our knowledge, this is the first time that an inherent resistance mechanism was conquered by a combination of a phage and an antibiotic. Given the genetic relatedness of phage SALSA and other phages, (e.g., SM9-3Y), it is likely that this beneficial antibacterial effect can also be achieved with those related phages. In order to test the robustness of the synergistic effect we tested additional antibiotic/SAM and phage concentrations. In a 16 h-infection assay, the synergistic effects could be confirmed with SAM and phage even at lowest tested concentration, i.e., 4 mg/L of antibiotic, and the MOI 10^−6^, respectively (green lines, Figure 4). Complete bacterial clearance was achieved throughout with the dual antimicrobial approach, in contrast to the single antimicrobial applications.

The same was true using MEM as synergistic partner; here the lowest tested concentration of 0.0625 mg/L and again the phage MOI 10^−6^ still enabled complete bacterial eradication after 16 h (green lines, Figure 5). 

When testing the strain SM04, synergistic effects between phage and SAM were also observed until a MOI of 10^−2^, and a minimum SAM concentration of 32 mg/L (Appendix A). Interestingly, the reduction of strain SM14 in liquid medium with phage alone was already very efficient, so that the addition of SAM started to improve the outcome only at a phage MOI of 10^-6^ (Appendix A). This phenomenon has been previously observed [7] and gives hope that, even when only a low proportion of phages might reach the site of infection during phage therapy, a co-administered antibiotic could sustain the antibacterial activity.

No synergistic effect was observed with SAM/SALSA against the two bacterial strains SM19 and SM20 (Appendix A). This indicates that a positive interaction with an antibiotic requires a true phage infection; the mere binding capacity of the phage does not exert enough stress on the bacteria (at least not with the phage concentrations tested). An antibacterial effect was also not observed for the strains SM10 and SM11 in liquid medium, neither alone, nor with the co-addition of the antibiotic. This is surprising, since the phage was able to produce clear plaques on solid medium indicative of productive phage infection. Apparently, the phage failed to replicate well in the liquid culture under the tested conditions, an effect which has also been observed in other studies [51,52]. In future experiments it would be interesting to assess whether a bacterial control is still possible when the strains are allowed to grow as a biofilm, which more closely mimics the environmental conditions of a true treatment for instance wound infections [53].

We have shown that a “contra-indicated” antibiotic and a T7-like phage engage with each other in eradicating clinical isolates of *Serratia marcescens*. The reasons for the synergistic interaction are unknown. However, it has recently been shown that the ampicillin-induced over-expression of AmpC goes along with changes in outer membrane profiles, which in turn can foster the efficiency of phage infection [54]. Although, this effect was demonstrated with a phage infecting *E. coli*, it is likely that the same mechanism has led to the observed synergy between ampicillin and phage SALSA. While, third- or fourth-generation cephalosporins are the standard treatment for *Serratia* infections, carbapenems have also been suggested as possible antibiotics of choice [55]. However, carbapenems have to be distributed carefully, not only because of their side effects such as secondary infections (e.g., mycosis) and neurotoxicity with the danger of inducing a seizure, but also because of the continuing rise of carbapenem-resistant gram-negative bacteria, including *S. marcescens* [56,57,58]. This positive interaction between phage SALSA and SAM implies that it is fundamentally possible to tackle the axiomatic condition of intrinsic drug resistance. The possibility of enhancing the spectrum of an a priori useless antibiotic, by combining it with phages without the need for any “last resort” antibiotics, provides a new way in the ongoing fight against increasing antimicrobial resistance. This concept could be extended to other clinically relevant bacteria, e.g., primary ampicillin resistance in *P. aeruginosa*, or primary cephalosporin resistance in *Enterococcus sp.* Clearly, a detailed look into host-phage interactions at molecular level is required for a better mechanistic understanding as well as in vivo studies to verify the excess therapeutic value of the combinational approach. Nevertheless, instead of phage-only concepts, we believe, that a dual antibacterial approach might represent the more promising approach for curing infections caused by multi-drug resistant bacteria in future.

## 3. Materials and Methods

### 3.1. Bacterial Strains and Phage Isolation

A total of 20 clinical isolates of *S. marcescens* (designated as SM01-SM20) were used in this study. Bacterial strains were grown in Lysogeny Broth-LB (NaCl 1% *w/v*, tryptone 1% *w/v*, yeast extract 0.5% *w/v*). Phage SALSA was isolated from sewage taken from the wastewater treatment plant Aachen-Soers, Germany, using strain SM01 for propagation. To this end, 5 mL of sewage was purified from bacteria by centrifugation (Eppendorf 5810R, Hamburg, Germany) at 2.330 × g for 10 min followed by sterile filtration of the supernatant using a 0.45-μm-pore-size and a 0.2-μm-pore-size sterile filter (Filtropur S, Sarstedt, Nümbrecht, Germany). The purified sewage was then mixed with 5 mL of 2× LB-medium, and 100 µL of the respective strain was added at an OD_600_ of 1. After overnight incubation at 37 °C and 200 rpm (VXR Vibrax, IKA-Labortechnik, Staufen, Germany), the mixture was centrifuged at 2330 × g for 10 min and filtered twice with a 0.45-µm-pore-size and a 0.2-µm-pore-size sterile filter. The phage-lysate was further purified by five consecutive single plaque isolations and propagations. The phage titer was determined as the number of plaque forming units [pfu/mL] by the Double Agar Overlay Plaque Assay, as described previously [49].

### 3.2. Characterization of the Clinical Isolates of S. marcescens

MALDI-TOF mass spectrometry (Microflex LT, Bruker Daltonik GmbH, Bremen, Germany) was used to verify species identity. Bacterial isolates were differentiated via enterobacterial repetitive intergenic consensus PCR (ERIC-PCR), as described previously [7]. The total volume per sample was 50 µL with each sample containing GoTaq Flexi Buffer, 4mM MgCl_2_, 0.2 mM dNTPs (Roche Applied Science, Penzberg, Germany), 2.0 μM of the forward primer (ERIC-1: 5′-ATGTAAGCTCCTGGGGATTCAC-3′) as well as 2.0 µM of the reverse primer (ERIC-2: 5′-AAGTAAGTGACTGGGGTGAGCG-3′), 2.5 U of GoTaq G2 Flexi DNA Polymerase and approximately 50 ng of template DNA.

The PCR-reaction started with an initial denaturation at 95 °C for 7 min, followed by 35 cycles of denaturation at 94 °C for 30 s. The primer annealing was done at 52 °C for 1 min, the elongation at 72 °C for 8 min and a further, final elongation step at 72 °C for 16 min. Afterwards, the DNA-fragments were run on a 1.5% TBE-agarose gel at 70V for 3h at room temperature and visualized via GelStudio SA System (Analytik Jena, Jena, Germany). Further analyzation of the DNA fingerprints was done using GelQuest and ClusterVis (Sequentix, Klein Raden, Germany).

### 3.3. Host Range

The host range of the phage for the 20 *S. marcescens* strains was evaluated via the spotting method [59]. To this end, 50 µL of the respective bacterial strain were plated onto agar and afterwards spotted with 5 µL of the phage at different dilutions (range from 10^8^ pfu/mL to 10^2^ pfu/mL). After overnight incubation at 37 °C, productive lysis by the phage was assumed in cases where individual plaques were visible within the spotting zone at appropriate phage dilutions. For the positive cases, efficiency of plating (EOP) [59] was determined relative to the propagations strain SM01.

### 3.4. Transmission Electron Microscopy (TEM)

With respect to TEM, the samples were prepared by exchanging a high-titer phage lysate in HEPES buffer via Amicon Ultra −0.5 mL Centrifugal Filter Units 100K (Merck, Darmstadt, Germany). The phages then adsorbed on glow discharged formvar-carbon-coated nickel grids (Maxtaform, 200 mesh, Plano, Wetzlar, Germany) for 10 min. Afterwards the samples were stained with a drop of 1% phosphotungstic acid (in aqua dest., adjusted to pH 7.2; Agar Scientific Ltd., Stansted, United Kingdom) and then air-dried before they were examined using a TEM LEO 906 (Carl Zeiss, Oberkochen, Germany), operating at an acceleration voltage of 60 kV. For photography, a wide-angle Dual Speed 2K-CCD-Camera 14 bit (Tröndle, TRS Moorenweis, Germany) and the analysis software IMAGE SP Professional (SISPROG, Tröndle, Moorenweis, Germany) were used. A description of phage morphology, and determination of the size of phage head and tails, were performed based on five selected virions displayed on the electron micrographs (average size ± standard deviation).

### 3.5. Phage Genome Sequencing and Analysis

Phage DNA isolation was carried out using the QIAamp DNA Mini Kit (Qiagen, Hilden, Germany) following the respective guidelines of the manufacturer (DNA Purification from blood and body fluids). Prior to the extraction of phage DNA, remnants of potential bacterial DNA were removed through a DNAse digest for 15 min. Whole genome sequencing was performed with the MiSeq platform (Illumina, San Diego, United States of America), as described in the product manual (Nextera XT DNA Sample Preparation Guide). For sequencing, a paired-end library was generated using the Nextera XT Library Prep Kit, and 2 × 150 bp reads were generated using the MiSeq v2 Reagent Kit. De novo assembly of reads was performed via the St. Petersburg genome assembler (SPAdes) [60]. The function BLASTn was used to search for closely related phage genomes publicly available in GenBank. GeneMark.S [61] was used to identify potential ORFs and annotation was performed using PHASTER [62]. Refined annotation was performed by comparing each ORF with the public database via the function BLASTp and by directly comparing each ORF with those of the closely related and well annotated phage YeO3-12 [26]. Entire-genome based phylogenomic treeing analysis with selected phages was performed with Genome-to-Genome Distance Calculator VICTOR based on the Genome-BLAST Distance Phylogeny method (GBDP) inferred using the formula D6 [25]. The genome nucleotide sequence of phage SALSA has been deposited to GenBank under the accession number MT419366.

### 3.6. Liquid Infection Assay

Liquid infection assays were performed with every strain that was permissive for productive lysis on solid medium and with the strains for which “lysis from without” [16] was observed. This study focused on the investigation of possible positive interactions between phage SALSA and ampicillin/sulbactam, but included meropenem for comparative reasons. All strains are intrinsically resistant against ampicillin/sulbactam but sensitive to meropenem. Bacteria were challenged with four phage concentrations (MOI 1, 10^−2^, 10^−4^, 10^−6^), either alone or combined with five different antibiotic concentrations ranging below and above the MICs determined for the propagation strain (i.e., the MIC of SM01 for ampicillin/sulbactam was 16 mg/L, thus the concentrations were ranging between 4 mg/L and 64 mg/L, accordingly). For comparison, the strains were also challenged with the antibiotics alone. Likewise, the concentration range of meropenem (MIC 0.25 mg/L) was selected between 0.0625 mg/L and 1.0 mg/L.

We inoculated 10 mL of 2× LB with 100 µL of the host strain and incubated overnight at 37 °C and 200 rpm. Subsequently, 5 mL of this culture were again mixed with 5 mL 2× LB and shaken at room-temperature for another hour to reach bacterial concentration of approximately 5 × 10^8^ CFU/mL. Phage-lysates were diluted by ten-fold serial dilution. 100 µL of phage-lysates were inoculated with 98 µL of bacterial strain and 2 µL of the selected antibiotic on a 96-well microtiter plate. For controls, 98 µL of the host strain or LB medium were mixed with 2 µL RNase-free water and 100 µL PBS. After sealing the microtiter plate with an adhesive tape, a hole was made in every well to supply the bacteria with oxygen throughout the experiment. The microtiter plate was then placed into the microplate reader SpectraMax i3 (Molecular Devices, Sunnyvale, United States of America) where the OD_590_ at 37 °C was measured every 20 min over 16h or 68h. All assays were run in triplicate. At the end of the infection assays, the samples were plated onto agar in adequate dilutions in order to determine the number of viable bacteria.

## Figures and Tables

**Figure 1 antibiotics-09-00371-f001:**
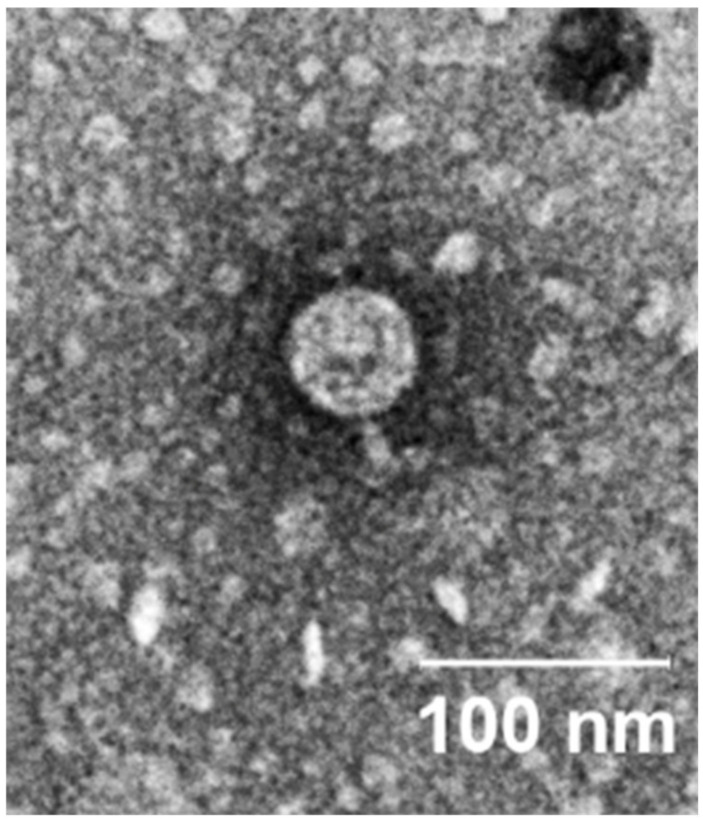
Electron micrograph image of phage SALSA infecting *S. marcescens*, negatively stained with 1% phosphotungstic acid.

**Figure 2 antibiotics-09-00371-f002:**
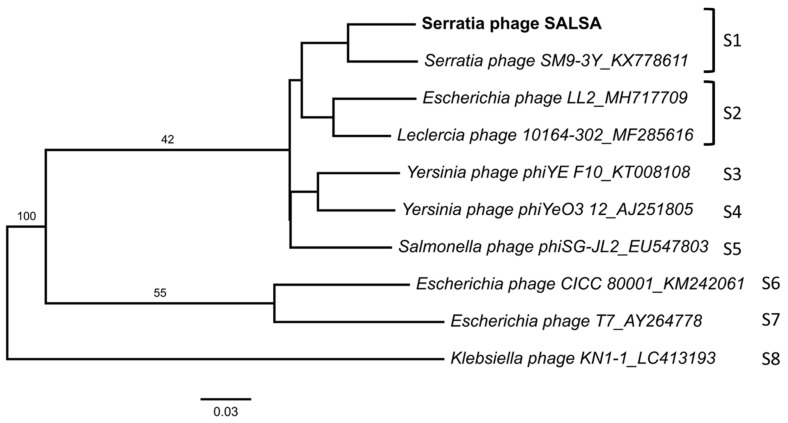
Whole genome-based phylogenomic treeing analysis, based on the Genome-BLAST Distance Phylogeny method (GBDP), is referred using the formula D6. The numbers above branches are GBDP pseudo-bootstrap support values from 100 replications. The branch lengths of the resulting VICTOR trees are scaled in terms of the respective distance formula used. The OPTSIL clustering yielded eight species clusters (S1 to S8).

**Figure 3 antibiotics-09-00371-f003:**
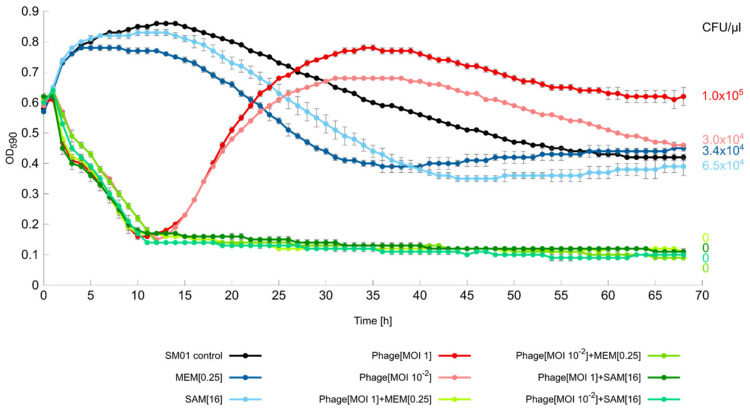
Liquid infection assays against *S. marcescens* strain SM01 with or without the antibiotic ampicillin/sulbactam (SAM, 16 mg/L) or meropenem (MEM, 0.25 mg/L). Reduction was measured via optical density at 590 nm (OD_590_). Each experiment was performed in triplicate and the means ± standard errors are indicated. The number of colony forming units/µL (CFU/µL) after 68h is given on the right side.

**Figure 4 antibiotics-09-00371-f004:**
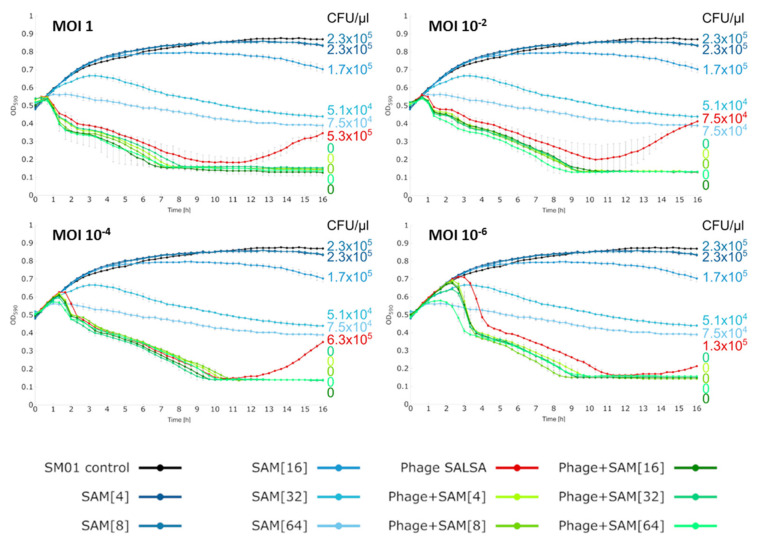
Liquid infection assays against *S. marcescens* strain SM01 with or without various concentrations (mg/L) of the antibiotic ampicillin/sulbactam (SAM). Reduction was measured via optical density at 590 nm (OD_590_). Each experiment was performed in triplicate and the means ± standard errors are indicated. The number of colony forming units/µL (CFU/µL) after 16h is given on the right side. Data are from one set of experiments but are shown in four graphs for better visibility of the phage/antibiotic synergy.

**Figure 5 antibiotics-09-00371-f005:**
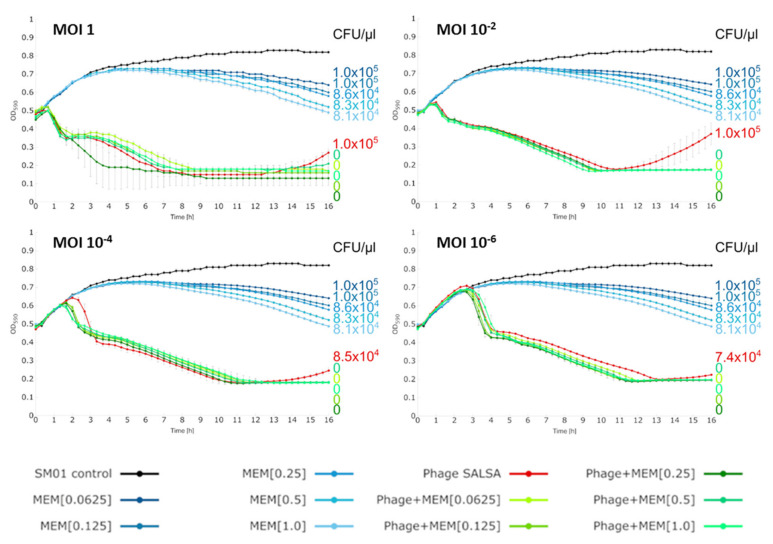
Liquid infection assays against *S. marcescens* strain SM01 with or without various concentrations (mg/L) of the antibiotic meropenem (MEM). Reduction was measured via optical density at 590 nm (OD_590_). Each experiment was performed in triplicate and the means ± standard errors are indicated. The number of colony forming units/µL (CFU/µL) after 16h is given on the right side. Data are from one set of experiments but are shown in four graphs for better visibility of the phage/antibiotic synergy.

**Table 1 antibiotics-09-00371-t001:** Genome wide nucleotide identity of SALSA with the five most closely related phages.

Related Phage	Query Cover	Identity	Accession-No.
*Serratia* phage SM9-3Y	95%	96.4%	KX778611.3
*Escherichia* phage LL2	88%	96.2%	MH717709.1
*Leclercia* phage 10164-302	89%	96.6%	MF285616.1
*Yersinia* phage phiYeO3-12	90%	96.3%	AJ251805.1
*Salmonella* phage phiSG-JL2	87%	96.3%	EU547803.1

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
