# Peer review of "Tackling Intrinsic Antibiotic Resistance in Serratia marcescens with a Combination of Ampicillin/Sulbactam and Phage SALSA"

_antibiotics, 2020, doi:10.3390/antibiotics9070371_

Round 1
Reviewer 1 Report
The manuscript by Weber et. al., describes and characterizes phage SALSA isolated against a clinical strain of Serratia marcescens, an increasingly relevant pathogen. Infections by S. marcescens poses a unique challenge to antibiotic treatments, as Serratia is intrinsically resistant to b-lactams due to the chromosomally encoded AmpC-type b-lactamases. Hence, from a clinical and therapeutic standpoint isolation any phage or antimicrobial is a significant step forward. SALSA was able infect and propagate to varying degrees in 5 strains out 20 clinical strains tested in this study and showed partial activity against two more strains. Morphologically SALSA is T7-like podophage with ~40 kb genome which shares significant (96.4% identity over 95% of the genome) sequence similarity with previously characterized Serratia phage SM9-3Y. The similarity also extends to other Enterobacteriaceae phages and most of the ORFs share >96% identity at the protein level with 4 to 9 ORFs sharing 100% identity. To test the combinatorial effect of SALSA with antibiotics, authors conducted series of tests experimenting with both the multiplicity of infection (MOI) and different concentration of ampicillin/sulbactum (SAM) and meropenem (MEM) antibiotics. Surprisingly, over 68 h combination of both the phage and antibiotics practically eliminated CFUs of Serratia strain SM01. Similar experiments over 16 h duration showed that low MOI (10^-6) and low concentrations of antibiotics are sufficient to eliminate most, if not all the CFUs. Overall, this is an interesting study exploring the synergistic effects of phage and antibiotics that were previously ineffective against the pathogen.
Major issues:
Given the high degree of similarity with SM9-3Y and other phages to SALSA, wouldn’t the expectation be that treatment with other phages might also sensitize bacteria to SAM and MEM? If so, please include that as part of the discussion.
Are the CFUs recovered after re-growth in phage-only treated conditions (Fig. 4) sensitive to SAM and MEM?
Line 112: The dimensions of the phages are described to the tenths of a nanometer. I doubt the measurement is accurate enough to discern to that level. Either round up the number to the nearest whole number or provide more information about the accuracy of the measurement.
In the final remarks comment about plausible reason for sensitization to antibiotics.
Minor issues:
Figure 1. Scale bar is not clear in the image. Change the color to white to make it more obvious over the grey background of the image.
Figure 3: as presented is not easy to interpret. Showing a gene synteny figure with all five T7-like phages would be more informative. Especially, to understand the genomic context of the highly conserved ORFs.
Line 163: Until recently?
Line 215: the word “enjoy” anthropomorphizes non-sentient microbes.
The order in which supplementary figures are cited in the text is not right S1 should come before S2 and S3 (Line 266-274).
Line 336: used spotted instead of inoculated.
Reviewer 2 Report
The manuscript describes noteworthy research that demonstrates the in vitro synergy between a single bacteriophage and antibiotics in the control of Serratia marcescens (Sm). Under Major Comments, the authors should clarify and/or further explain the important issues raised during the review process. Overall, this is well executed research that was interesting to read.
Major Comments:
L67-70 The introduction should leave the reader with a clear idea of what the researchers are trying to achieve. The end of the introduction should have a paragraph or a couple of sentences that clearly state the research objectives and hypothesis. This is lacking in this manuscript.
L69 “The data show …” This sentence is a conclusion and it should be moved to the results/discussion.
L75 What locations, geographical or spatial did these 20 isolates originate from? The reader should not have to go to the supplement for this basic information. You mention that the 20 Sm isolates were placed into 8 groups based on their sensitivity to antibiotics. Once again, the reader should not need to wander through supplementary material to learn what antibiotic sensitivity groups S. marcescens SM01, SM04, SM10 and SM14 fall into since these specific bacterial isolates are the core of the study.
L83 Phenotypic characterization of phage SALSA.
General side comment to the authors: what surprises me here is that only a single Sm isolate was used to screen the sewage and that you zeroed in on the single Podoviridae phage. Why not isolate on a mixture of bacterial Sm isolates? In addition, were you following the old bacteriophage directive that only clear plaques produce lytic phage with good efficacy against pathogens, while cloudy plaques indicate presence of lysogens? Scientific literature shows that this “directive” does not hold in the phage infection of Enterobacteriaceae?
Now to the main question for this section – In your isolation from sewage, why did you want to specifically isolate a single Podo- phage? You did mention that Podos are not abundant in studies (L187-188)? Did you have more than one Podo- phage that you tested against Sm? Any other families of Sm phage in that sewage?
L93 “low host range” replace with “narrow host range”
L97 “Low host range …..limitations in phage therapy” Narrow host range may be advantageous in certain pathogen/host combinations. You know very little at about the antibiotic resistance in SM and phage receptors. This general statement should be avoided or further explained.
L91-98 I do not understand this entire section. How did you determine “lysis from without” for strains SM19 and SM20? Why focus only on this mechanism as a possible explanation. There are other possibilities/theories as to what may be occurring.
L98 In this work the bacteriophage host range was determined by the spot test. The most primitive of methods (refer to Abedon published manuscripts). Therefore, I would highly recommend that the authors avoid conclusion on the broad vs. narrow host range of SALSA when it is based on a spot test. Did you test any of the related bacterial genera?
L100-108 Why in the world are you discussing “phage culturomics” before you consider phage cocktails? Realistically, you would never use a single phage in any type of therapy. This whole section needs a rewrite – You used a single phage/antibiotics to demonstrate synergy. What is unknow is whether a phage cocktail and antibiotics would improve efficacy. The phage cocktail, would certainly avoid the development of bacterial resistance to SALSA during therapy.
L119 Genomic analyses of SALSA: Table 1 (5 related phage), Figure 2 (whole genome) and Figure 3 (protein level). These figures/table show the same data in different ways. Two should be moved to the Supplement material.
L217-235 In this long treatise on problems with bacterial regrowth, I find it strange that the authors do not mention the use of phage cocktails in phage therapy, which may result in better efficacy and avoid the development of resistance. Single phage and even a phage cocktails, generally do not clear a bacterial liquid culture. Do not focus on this point, since your manuscript shows elegantly that the actual bacterial population in the phage/antibiotic combinations reduced to zero.
L388 Liquid infection assay - The OD590 of 0.6 and 0.5, what does it equates to in terms of CFU/µl or the better more commonly used CFU/ml? It would be interesting to the know the starting concentration of bacteria and phage in the initial 1 µl or 1ml. I cannot find this in the materials and methods or on the graphs.
L405 to end of manuscript References need to be checked. Many of them have manuscript titles with all capitalised words (References #2, 3$, #6, #9 etc.) and bacterial Latin names that are not italicised. Entire section should be carefully checked and brought to a standard format.
Minor Edits/Suggestions:
L35-38 “Despite some ……phage therapy.” This sentence needs to be rewritten with a clear subject. What do you mean by “despite desirable features” or is the subject phage resistance?
L40-43 Another set of awkward sentences. Please rewrite and clarify what you wish to describe.
L47 “ineffective antibiotic” What is the definition? Ineffective at certain concentrations? Ineffective due to development of antimicrobial resistance? What is a “suitable phage”?
L53 Delete “For instance” Start sentence with “The chromosomal…..”
L56 Delete words “long time”
L87 Initially sewage filtrate was spotted onto a SM01 bacteria lawn and the clear spots were used for phage isolation. What is meant by “further titration”? I think you mean dilution of the liquid obtained from the spots once the agar was removed.
L114 TEM enlarge the phage image. Micron marker should be at 50 or 100 nm (at the most).
L215-216 “…..except for the phage attack, bacteria enjoy conditions in full …” This is a good example of anthropomorphism in scientific writing.
L238 Delete “and strongly”
L273-274 Any other reason why synergy did not occur? The lysis from without is not a convincing conclusion.
L275 “… requires the permissiveness of the bacteria.” What does this mean?
L279-281 Confusing explanation of phage lysis of these isolates on solid vs liquid media. What you mean by “not permissive”. Perhaps with certain phage-host combinations you require a critical number of phage particles in a solution for infection to occur?
L307-315 The Materials and Methods in this section are written in such a manner that no one could repeat these protocols. Details are needed centrifugation (type), sterile filtration (this can be done in many ways?), LB medium (Luria Bertani?, manufacturer?, concentration?), what does 2X LB mean?, shaker (they come in many types), 45 um filter manufacturer? Place?
Round 2
Reviewer 1 Report
The authors have satisfactorily addressed the concerns that I had raised in my review and I don't have any issues.